# Mucin and Agitation Shape Predation of *Escherichia coli* by Lytic Coliphage

**DOI:** 10.3390/microorganisms11020508

**Published:** 2023-02-17

**Authors:** Amanda Carroll-Portillo, Kellin N. Rumsey, Cody A. Braun, Derek M. Lin, Cristina N. Coffman, Joe A. Alcock, Sudha B. Singh, Henry C. Lin

**Affiliations:** 1Division of Gastroenterology and Hepatology, University of New Mexico, Albuquerque, NM 87131, USA; 2Statistical Sciences, Los Alamos National Laboratory, Los Alamos, NM 87545, USA; 3Biomedical Research Institute of New Mexico, Albuquerque, NM 87108, USA; 4Department of Emergency Medicine, University of New Mexico, Albuquerque, NM 87131, USA; 5Medicine Service, New Mexico VA Health Care System, Albuquerque, NM 87108, USA

**Keywords:** bacteriophage, lytic, mucin, gastrointestinal tract, motility

## Abstract

The ability of bacteriophage (phage), abundant within the gastrointestinal microbiome, to regulate bacterial populations within the same micro-environment offers prophylactic and therapeutic opportunities. Bacteria and phage have both been shown to interact intimately with mucin, and these interactions invariably effect the outcomes of phage predation within the intestine. To better understand the influence of the gastrointestinal micro-environment on phage predation, we employed enclosed, in vitro systems to investigate the roles of mucin concentration and agitation as a function of phage type and number on bacterial killing. Using two lytic coliphage, T4 and PhiX174, bacterial viability was quantified following exposure to phages at different multiplicities of infection (MOI) within increasing, physiological levels of mucin (0–4%) with and without agitation. Comparison of bacterial viability outcomes demonstrated that at low MOI, agitation in combination with higher mucin concentration (>2%) inhibited phage predation by both phages. However, when MOI was increased, PhiX predation was recovered regardless of mucin concentration or agitation. In contrast, only constant agitation of samples containing a high MOI of T4 demonstrated phage predation; briefly agitated samples remained hindered. Our results demonstrate that each phage–bacteria pairing is uniquely influenced by environmental factors, and these should be considered when determining the potential efficacy of phage predation under homeostatic or therapeutic circumstances.

## 1. Introduction

Bacteria are ubiquitous in nature, occupying every biome, including aquatic environments, soil, and other living creatures (plants, insects, animals, etc.). Perhaps even more prevalent are the viruses that infect these bacteria, bacteriophages (phages) [1]. Infection of bacteria by phages can take many forms, including aggressive action (bacterial lysis) or integration of the phage genome into the bacterial DNA (prophage domestication) [2]. Use of phages as treatment for bacterial infections, phage therapy, has long been studied, but there is renewed interest due to increased prevalence of antibiotic resistance and a wide range of potential uses for bacteriophages [3,4,5,6].

Harnessing phage–bacteria interactions to our advantage either to manipulate bacterial populations or as therapy for a perturbed gut microbiome (dysbiosis) requires a better understanding of how these interactions occur. Akin to their mammalian counterparts, bacterial viruses initiate infection through receptor-mediated interactions between receptor binding proteins (RBP) on the viral surface and their cognate receptor(s) on the bacterial surface. RBPs convey the level of specificity each phage has for its bacterial host with most phages remaining fairly confined to a specific type of bacteria [7,8,9]. However, both exchange of RBP sequences that could occur during horizontal gene transfer events and promiscuous phages have been reported, allowing for one type of phage to differentially regulate several types of bacteria within a community [7,10,11,12]. 

The external factors of the surrounding micro-environment also influence the occurrence and outcomes of bacteria–phage interactions. The gastrointestinal micro-environment is a perfect example where physiochemical factors (pH, ion concentration, fluid flow, etc.) [13,14] combined with host factors (intestinal motility, neutralizing antibodies, cytokines, bile acids, etc.) [13] influence the regulation of bacterial populations by cognate phages in eubiosis and dysbiosis [15]. Chibani-Chennoufi et al. [16] demonstrated that the effectiveness of bacteriophage within the gut was not solely driven by increasing phage numbers. They found that administration of lytic coliphage to mice resulted in an expected increase in phage number isolated from fecal material with a concurrent expected decrease in *Escherichia coli* (*E. coli*). However, *E. coli* was never completely removed from the system; instead, the bacteria reached a stabilized number despite still being susceptible to administered phage. While external factors likely have a combinatorial effect, within the gastrointestinal tract, mucin has the propensity to have an outsized impact on phage–bacteria interactions.

Mucin serves to protect the intestinal mucosa from prolonged contact with commensal organisms as well as from invading pathogens through both its physical characteristics and the support it offers to a gradient of antimicrobials [17,18,19,20]. The composition and concentration of mucin varies across the length of the gastrointestinal tract with different mucin proteins dominating different regions [21,22]. Within the small intestine, Muc2 is the dominant gel-forming mucin released by goblet cells found along the epithelium [22,23]. In conjunction with several transmembrane mucins, Muc2 forms a loose, net-like structure that resides on top of the epithelium to aid in hindering contact between the cellular surface and the bacteria residing in the lumen. Due to its loose attachment, the mucin of the small intestine sloughs, and it and all its contents move distally with normal gastrointestinal motility [23]. In contrast, mucin of the colon forms two layers, an inner, dense layer residing above the epithelium and an upper, more porous layer [22]. Both serve to protect the colonic epithelium from bacterial invasion. 

Given that bacteria are known to adhere to mucin [24] and phages also have affinity with accumulation in mucosal secretions [9,25,26] it is unsurprising that there are several reports of phage–bacteria interactions influenced by their surrounding mucin micro-environment [26,27,28]. The nature and consequences of bacteria and phage interactions with mucin was recently reviewed by Rothschild-Rodriguez et al. [29]. Agitation could further affect these interactions through its impact on mucin. While the physiological concentration of mucin in the small intestine has been reported to vary between 2 and 5% [30], several types of gastrointestinal dysbiosis are reported to result in either a decrease [19,31] or an increase in mucin [32,33]. Fluctuations of bacterial populations [34,35] and phage populations [25,36,37,38,39] are also reported in dysbiotic conditions associated with gastrointestinal diseases.

To examine the influence of mucin on the effectiveness of phage predation, we employed a simplified system using *E. coli* and two different lytic coliphage (T4 and PhiX174) in increasing concentrations of mucin. Our results suggest that bacterial motility, agitation, and phage number all play a role in driving the outcomes of bacteria–phage interactions within a mucin micro-environment. 

## 2. Materials and Methods

### 2.1. Bacteria and Bacteriophage

Wildtype *E. coli* (K803), a K-12 derivative lacking prophage lambda [40], served as the host bacterium in all experiments. Bacteria were grown in Luria Bertani (LB) broth at 37 °C overnight with shaking (~225 rotations per minute (rpm)). T4 and PhiX174 (PhiX) coliphages were purchased from Carolina Biological (Burlington, NC, USA) and ATCC respectively. Both phages were propagated on the K803 background. T4 is a *Caudovirales* phage and has been demonstrated to have affinity for mucin through its Hoc capsid protein [26] while PhiX is a *Microviridae* phage with no known mucin-affinity Ig-like domains (IgBlast; [41]). Briefly, bacteria were grown to mid log phase in LB and phages were added at a 1:10 ratio. Cultures continued shaking at 37 °C for another 4–5 h, after which phages were harvested. Bacteria were pelleted at 3500× *g* for 20 min. Supernatants were collected and filtered with 0.45 μm syringe filters. Confirmation of phage numbers for all assays was performed by quantifying phages using soft agar overlay assay for the counting of plaque forming units. Bacterial numbers were quantified using the QUANTOM Tx Microbial Cell Counter (Logos Biosystems, Annandale, VA, USA) with counts verified through plating. 

### 2.2. Mucin

Porcine stomach mucin, Type II (Sigma, St. Louis, MO, USA) was used at different concentrations. Stock solution of 10% mucin in LB was stirred until well mixed and the solution pH was brought to 7 with sterile 1N NaOH. Stock solution was then brought to a boil to clear possible contamination and stored at 4 °C. Boiling was not detrimental to the effect of mucin, similar to previous findings [42]. Prior to assays, stock solution was diluted to the desired concentration (0.5, 1, 2, 4%) in LB and mixed on a nutator at room temperature overnight.

### 2.3. Plate Motility Assays E. coli 

Motility in the context of increasing mucin concentrations was assessed using plate motility assays. Motility plates were made the day prior to assay by mixing phages with mucin, 2× LB, water, and 1X Tetrazolium chloride (0.005%, Sigma, St. Louis, MO, USA) within each plate. To achieve final desired concentrations of mucin (0%–4%), an appropriate amount of 10% mucin stock solution was added. Plates were swirled briefly to mix contents, and 2× motility agar solution (final concentration of 0.27% agar in LB) was added. Plates were allowed to solidify overnight at 4 °C and equilibrated to room temperature the day of the assay. Phages were added at 10^4^ plaque forming units (pfu)/plate (1:1 bacteria:phage ratio), 10^3^ pfu/plate (1:0.1 bacteria to phage ratio), or 10^2^ pfu/plate (1:0.01 bacteria:phage ratio). The following day, K803 bacteria (5 μL, 10^4^ bacteria) were inoculated into the center of solidified plates. Plates were allowed to settle for one hour prior to incubating at 37 °C overnight. Bacterial motility was measured as distance of observed red color (an indicator of bacterial metabolism of Tetrazolium) from inoculation point.

### 2.4. In Tube Assays

Phage predation as a function of mucin concentration was quantified using bacterial growth following exposure to phage as a readout. Static (no mixing) and dynamic (with mixing) conditions were compared. *Static:* One mL samples of varying mucin concentration were prepared in 1.5 mL microcentrifuge tubes. Phages were added at MOI of either 1 (10^3^ pfu) or 100 (10^5^ pfu) and mixed with inversion. Dilutions of overnight K803 cultures were made in LB and allowed to sit at room temperature for 30 min to allow for initiation of log phase growth. To each tube, ~10^3^ bacteria were added in a 10 μL volume without mixing. Samples were incubated for an additional 3 h, after which 100 μL was withdrawn for plating without any sample mixing or further dilution. *Dynamic:* Dynamic, in-tube assays were performed to assess the effects of either brief or constant agitation on the effectiveness of phage predation within mucin-containing environments. Experimental set up was similar to static assays with minor variations. Brief agitation was achieved by mixing samples with inversion at three different times: with addition of phages, with addition of bacteria, and once more prior to plating. Constant agitation involved incubating samples in a 37 °C shaker at 100 rpm for the entire 3 h incubation period prior to plating for bacterial quantitation. This rpm was on par with previously described in vitro methods to mimic small intestinal mixing [43]. Inclusion of mucin free samples containing bacteria and phages at either MOI of 1 or 100 in LB, ensured that lysis of bacteria was occurring in samples as expected (see Appendix A).

Quantitation of bacterial growth from all in-tube assays was performed using FIJI software, v2.3.0/1.53f51; Java 1.8.0_202 [44]. Photos of plates were taken after overnight incubation at 37 °C and images were processed. Colony features varied greatly on plates from samples containing phages (see Appendix A). To remove bias as to which colonies should count, overall bacterial growth was instead assessed to examine differences between samples. To do this, a large circular area covering most of the plate (region of interest, ROI) was drawn and its area was measured using the “Measure” function. The same ROI was used for all plates within the same assay. Each ROI was converted to eight bit, and regions of bacterial growth were thresholded with the Shanbhag feature. The “Analyze Particles” function was used (size set to “0-infinity” and circularity of 0.00–1.00) to measure the area of bacterial coverage within the defined circle for all plates. Areas of bacterial coverage were converted to percentage of total area (that of the ROI). Values were normalized to bacteria only control (in LB with no phage) to determine the fold difference in viable bacteria between the control and each of the samples.

### 2.5. Generation of Sobol Plots

Variance-based sensitivity analysis was conducted using the Sobol method [45,46,47] to decompose the variance of phage predation into fractions (called Sobol indices), which can be attributed to predictor variables (Agitation level, % Mucin, MOI) or sets of predictor variables. Sobol indices were computed for the cases described in Section 3.7 using the approach of Francom et al. [45], which uses Bayesian multivariate adaptive regression splines to model the (possibly non-linear) relationship between phage predation and the other predictor variables. Posterior distributions of the Sobol indices were computed using the BASS package in the R programming language (version 3.6.3) [48,49,50] and are presented as boxplots.

## 3. Results

### 3.1. Effect of Mucin Concentration on Bacterial Motility 

We assessed bacterial viability following phage predation under different conditions using the motile *E. coli* strain, K803 [40], and two lytic coliphages as predators, T4 and PhiX. Joiner et al. [51] have suggested that bacterial motility is the strongest determinant in the efficacy of phage predation. However, mucin has been shown to hinder bacterial motility [51,52] indicating that phage predation within the mucin-containing environment of the intestine might be detrimentally affected by loss of bacterial motility. To first assess the effect of increasing concentrations of mucin on K803 motility, we used a plate-based motility assay (Appendix A). After overnight incubation of centrally inoculated motility plates of increasing mucin concentrations, bacterial motility from the inoculation site was visualized using the chromogenic byproduct of Tetrazolium as a readout of where bacterial metabolism occurred. Comparison of the distances of red color beyond the original inoculation sites on motility plates (Appendix A) showed that increasing concentration of mucin decreased bacterial motility, as expected with measurements from the inoculation site to the region of outermost motility and shown in Table 1. By increasing mucin concentration to just 0.5%, the bacterial motility decreased by approximately 20%. Higher concentrations of mucin resulted in even more pronounced motility reduction, with an approximate 64% decrease in distance swum in 4% mucin. This substantial decrease in motility is further evidenced by the relatively small p-values from Dunnett’s multiple comparisons test (Table 1).

### 3.2. Effect of Phages on Ability of Bacteria to Spread

Addition of either T4 or PhiX to motility plates allowed for the examination of phage predation in the context of bacterial motility. The motility plates were most representative of a static environment where bacterial motility would be the major movement through the matrix and phage diffusion, if it occurred, would have been negligible [51,53,54]. Motility plates containing increasing concentrations of mucin (0%–4%) were infused with decreasing pfu of either phage for final bacteria:phage (B:P) ratios of 1:1, 1:0.1, and 1:0.01 (Figure 1). 

Images of bacterial expansion within the motility plate (indicated by red color) under each tested condition demonstrated that even when bacterial motility was hindered in the highest concentration of mucin (4%), phages contributed to increased hinderance of bacterial spread at low bacteria:phage ratio (<1:0.1). Bacteria were only able to overcome phage effects at the lowest ratio tested (1:0.01) with breakthrough bacterial motility occurring in both T4 and PhiX containing plates. At this lowest bacteria:phage ratio, distance of bacterial spread was approximate to that found in the no phage controls regardless of which phage was present. 

### 3.3. Effect of Mucin Concentration on Bacterial Growth

Experiments to quantify the level of phage predation by each phage under varying conditions required assurance that mucin concentrations used in our experimental conditions were not contributing to bacterial growth, as mucin has been shown to be a metabolic substrate for non-commensal *E. coli* [55,56,57]. As such, we quantified bacterial growth in mucin-containing samples (0.5%–4%) compared with growth in controls (LB alone) after incubation for 3 h. For in-tube experiments, experimental design, as described in the Materials and Methods section, involved plating 100 μL from a 1 mL sample volume where 10^3^ bacteria were used for seeding. As seed bacteria were diluted from overnight culture and allowed to incubate 30 min prior to introduction of phage followed by the 3 h experimental incubation time, it was likely that bacterial samples at the termination of the experiment were in early log phase. Colony counts on control plates (from LB alone samples) were consistent with this premise as numbers ranged from 3 × 10^3^ to 5 × 10^3^ at the end of the experiment compared with 100 to 200 at the initiation of the experiment (data not shown). To demonstrate that mucin did not affect bacterial growth within this time frame, values found for bacterial growth in mucin were normalized to controls (set at 1 and represented by a dashed line; Figure 2) and plotted as a function of fold change. These measures demonstrated that the tested mucin concentrations did not systematically enhance bacterial growth (values in Table 2) as compared with the control media (ANOVA *p*-value = 0.63).

### 3.4. Phage Predation at MOI 1 in a Static System

Motility plates only allowed for analyses of bacterial spread in the presence of phage rather than assessing effects of phage predation on bacterial viability. To determine the outcomes of phage predation as a function of mucin concentration, we used in-tube assays to assess bacterial survival following phage exposure by plating a small volume from each assay for bacterial outgrowth as described in the Materials and Methods section. Results from phage-containing samples were compared with the equivalent no-phage samples. Observation of bacterial colonies on the outgrowth plates revealed frequent occurrence of varied colony morphology in bacteria taken from phage-containing samples (Appendix A). Simple visualization of plates demonstrated substantial differences between conditions in the samples (Appendix A), however, use of colony counting software for colony forming units (CFU) did not reflect these differences as punctate colonies were not differentiated from large, healthy colonies. We therefore determined quantitation of area of bacterial growth to be a better metric to reflect our observations, and used the method described in the Materials and Methods section to allow for direct comparisons between samples.

We first quantitated bacterial survival as a function of mucin concentration in static systems where bacterial motility was the dominant form of microbial movement (similar to those represented in the motility plates). Phages were added to in-tube, static assays and incubated for a shorter duration of 3 h to more closely represent potential periods of interaction within the small intestine [58]. Bacterial growth after incubation with MOI 0.1 of either T4 or PhiX was quantitated and graphed as the fold change in area of bacterial growth from phage-containing samples as compared with that from the no phage, no mucin controls (Appendix A). As these numbers reflected no bacterial killing in any of the conditions tested, we opted to examine the effect of phages at an MOI of 1, a fold higher than the reported physiological MOI in feces [59,60]. 

After application of both phages at MOI 1 to the static system, fold change in area of bacterial growth normalized to controls (no phage, no mucin; represented as a dashed line at 1) was plotted for each condition tested (Figure 3, black bars). Increased concentrations of mucin (2% and 4%) resulted in improved bacterial killing by both phages as indicated by bars further below the dashed line. T4 (Figure 3A) and PhiX (Figure 3B) means ± SEM for each mucin concentration are shown in Table 3; one-way ANOVA p-values were 0.124 and 0.15 respectively.

### 3.5. Effect of Agitation on Phage Predation at MOI 1

Bacteria–phage interactions in static systems (motility plates and in-tube assays) demonstrated decreased bacterial motility and lower bacterial viability (increased phage predation) at higher concentrations of mucin (2% and 4%) for both phages tested. However, the micro-environment of the small intestine is not static in nature, with both “flushing” motility associated with phase three of interdigestive motility (intestinal housekeeper wave) that occurs between meals [61] and fed motility that is non-peristaltic and more consistent with mixing [62]. Furthermore, within such active fluidic environments, phages without affinity for mucin are not retained as well as those with mucin affinity suggesting a loss in opportunity for phage predation to occur by these phages [63]. To examine the consequences of mucin on phage predation within a “mixed” mucin-containing environment, we applied either brief or continuous agitation to our in-tube assays during the incubation period. Samples were prepared as described in the Materials and Methods section, with phage mixed thoroughly into the mucin prior to the addition of ~10^3^ bacteria. For brief agitation, samples containing bacteria and phage were briefly mixed by inversion prior to incubation and only again prior to plating. For continuous agitation, samples were incubated in a shaker at 100 rpm. Bacterial viability after treatment with an MOI 1 of either T4 or PhiX was again measured using fold change in area of bacterial growth across a range of mucin concentrations (0–4%) as compared with controls (no mucin, no phage). We found that the application of even brief agitation in the presence of high mucin concentration (≥2%) for both phages substantially increased bacterial viability (Figure 3A,B respectively). Comparison of the mean values for both phages after agitation in high mucin concentration with those found in the equivalent static mucin samples (Table 4) showed brief agitation with T4-containing samples resulted in 1.8- and 1.5-fold increased bacterial survival in 2% and 4% mucin, respectively, demonstrating a decrease in effective phage predation. Constant agitation caused 1.7- and 1.6-fold increased viability. Similarly, brief agitation of PhiX-containing samples resulted in 1.4- and 1.7-fold increased bacterial survival in 2% and 4% mucin, respectively, whereas constant agitation showed 1.3- and 1.7-fold increase.

Mucin concentrations of 2 and 4%, where shifts in phage predation were seen upon the addition of agitation, are closely aligned with concentrations reported within the small intestine [30]. Limiting our analysis to these cases (2–4% mucin, MOI 1) meant that we detected no meaningful difference between T4 and PhiX, as seen in Figure 3C. A notable increase in phage predation is seen, for both phages, only in the static environment. These claims are supported by the formal analysis of a linear model fit (two-way ANOVA), where the p-value for the relationship between phage predation and phage type is 0.396 and the p-value for the relationship between predation and agitation level is $1.88\times 10^{-8}.$

### 3.6. Phage Predation and Agitation in a Phage Therapy Setting

As lower levels of phage (MOI 1) were insufficient to control their bacterial hosts in higher concentrations of mucin once agitation was introduced into the system, we questioned whether an infusion of the same phage at levels consistent with those used for phage therapy applications (MOI 100) would be sufficient to overcome the hinderance on phage predation. We ran in-tube assays as before, testing across mucin concentrations under static and agitated (brief and constant) conditions (Figure 4). In the static systems (black bars), both T4 (Figure 4A) and PhiX (Figure 4B) at MOI 100 demonstrated higher levels of phage predation (less viable bacteria) than those seen at MOI 1 for the same phage (Table 5).

Once agitation was introduced into the system, unlike at MOI 1, the results for each phage were different. T4 still demonstrated hinderance of phage predation at 2% and 4% mucin concentrations under brief agitation (Figure 4A, gray bars; Table 6) with 2.4- and 2.1-fold decrease, respectively, in phage predation as compared with the static system. When constant agitation (white bars) was applied, there was some recovery of phage predation, although not completely to static levels, with only 1.6- and 1.4-fold decreases in predation as compared with static conditions. PhiX phage predation was more effective at MOI 100 regardless of mucin concentration or level of agitation with little variation across results (Figure 4B). Thus, phage numbers of PhiX were able to overcome constraints on phage predation that may be due to mucin concentration or agitation, however this was not the case for T4. Levels of bacterial growth in non-mucin-containing samples of PhiX were comparable to those found in mucin-containing samples suggesting that effectiveness is more due to the nature of the phage rather than its lack of mucin affinity. The enhanced nature of PhiX phage predation over that of T4 phage predation at MOI 100 under all conditions tested is obvious when bacterial growth after exposure to each phage is directly compared (Figure 4C).

### 3.7. Mucin and Agitation Influence Phage Predation in a Phage- and MOI-Specific Manner

To understand the complex relationships between phage predation of bacteria and the gastrointestinal environment, and how these relationships differ between the PhiX and T4 bacteriophages, we conducted a variance-based sensitivity analysis using Sobol’s method [45,46,47,64]. Sobol’s method is similar to analysis of variance in that it quantifies how much of the total uncertainty in a model is accounted for by each variable or its specific interactions. The advantages of using Sobol’s method include (i) the ability to capture complex non-linear relationships between variables and (ii) the way in which a clear understanding of the relationship between experimental factors and the response variable becomes possible and readily quantifiable (with uncertainty) [45]. Sobol indices can be used as a measure of the impact that different variables (agitation level, MOI, mucin concentration) will have on a response variable (bacterial viability following phage predation). The variance of bacterial viability, as represented by fold change in the amount of bacterial growth after exposure to phages relative to growth in controls, is deconstructed and attributed to each of the experimental variables and sets of variables. A Sobol index corresponding to each variable (or set of variables) was estimated and gives the proportion of bacterial viability after phage predation which can be explained by that variable (or set of variables) alone. The “total sensitivity” of a particular variable is obtained by summing together each of the Sobol indices which correspond to it. These “total sensitivity” scores are not interpretable as proportions but can be used as a proxy for the importance of each explanatory variable (averaged over the entire data range for that particular variable). 

For both the T4 and PhiX phages, 95% (and 50%) credible intervals for all Sobol indices and “total sensitivities” are plotted in Figure 5. The difference in the behavior of PhiX and T4 is immediately and markedly apparent. As seen in Figure 5A,B, nearly all of the variance seen in phage predation for PhiX can be attributed solely to MOI (Sobol index 95% CI: (089, 0.97)). The Sobol indices for “agitation level” and the interaction “agitation level × mucin concentration” were the next largest, with estimated values of 0.017 and 0.013 respectively, indicating that all other experimental variables explained very little of the variation in bacterial viability after PhiX phage predation. In contrast, bacterial viability following T4 phage predation has its variance attributed to many different sources. The “total sensitivity” scores for T4 suggest that the most important variable for explaining bacterial viability outcomes is agitation level, followed by mucin concentration, then by MOI, although all three of these factors had a sizeable influence on the effects of phage predation. Additionally, Figure 5C demonstrates that the interaction between agitation level and mucin concentration has a sizeable impact on bacterial killing by T4 (Sobol index estimate: 0.198, 95% CI: (0.13, 0.27)). That is, roughly 20% of the variation in the data for bacterial viability after T4 predation can be explained through this interaction (and not by agitation and mucin concentration on their own). Other interactions (mucin and MOI, agitation and MOI, or all variables together) did not contribute substantially to the variance in viability following phage predation. Total sensitivity scores for both T4 and PhiX are reported in Table 7. 

## 4. Discussion

To expand our understanding of phage predation of bacteria in the presence of mucin within the small intestine, we have used a deconstructed model that is a simplified, closed system with two different coliphagic phages and wildtype *E. coli*. Previously, Cremer and Arnoldini et al. [65,66] proposed that a majority of bacterial growth occurs in a “growth zone” within the proximal colon suggesting that bacteria within the small intestine might either be in stationary phase or an earlier log phase of growth. As phage predation requires bacterial growth, we opted to incorporate early log phase bacteria into our experimental model to investigate the possible outcomes of phage–bacteria interactions as a function of mucin concentration, phage number, and agitation. We found that the effectiveness of phage predation by each of these lytic coliphage, as measured by bacterial survival following phage exposure, differed. Of the variables tested, MOI and agitation were the dominant determinants of bacterial viability.

The plate motility assays demonstrated that, in a static system, as mucin concentration increased (while still being kept within physiological levels), bacterial motility decreased such that at 4% mucin, bacterial motility was severely limited consistent with previous findings [51,67]. When phages were then applied to these plates in a uniform manner, they were capable of further controlling bacterial motility beyond the inoculation site at a very low bacteria:phage ratio. 

While the static motility plate system informed bacterial spread into an environment uniformly populated by corresponding phages, it did not clarify the outcome of phage predation. Instead, we used in-tube assays that were specifically designed to characterize the efficacy of phage predation by each phage under different conditions through quantitation of bacterial viability following phage predation. Determination of bacterial motility required conversion of Tetrazolium to a visible chromogen within the motility plates, and thus, longer incubation periods (>8 h). Use of a similarly lengthy incubation period with in-tube assays presented several issues. First, mucin settled and required mixing prior to sample removal, which for static assays could potentially skew counts as bacteria could then be more exposed to phage during the mixing. Second, incubation of ≥6 h resulted in complete bacterial killing in several samples preventing any meaningful comparison of the effectiveness of the phages under varying conditions. Third, within the mucin of the small intestine, the effects of phage–bacteria colocalization and interaction are shorter lived as loosened mucin is periodically moved distally due to gastrointestinal motility with transit of the GI tract at approximately 2–4 h [68]. With these considerations in mind, in addition to the decided upon bacterial phase of growth (early log phase), we chose to incubate in-tube assays for a 3-h period to allow for analysis of outcomes pertinent to the small intestinal environment. An MOI of 0.1 for either phage did not adequately kill bacterial content in samples within this time period, so an MOI of 1 was tested instead. Additionally, an MOI of 100 was tested to more closely mimic phage therapy conditions.

Concepts of bacterial motility, phage diffusion, and mucin concentration were recently reviewed [29], and if bacterial motility was the only determining factor for successful phage predation, it would be expected that predation would decrease as motility was lost within increasing mucin concentrations. This is not what the static, in-tube assays showed. Instead, predation by both phages did not vary across low mucin concentration but became more effective at ≥2% at low MOI, and not much variation across all concentrations at high MOI. What did obviously change relative to mucin concentration was that predation by PhiX when compared with T4 at high MOI was more effective regardless of mucin concentration or agitation. These data suggest that while bacterial motility likely plays a role in phage predation in a static mucin environment, it is not necessarily the dominant variable. Phage type also plays a role.

While static assays allow for examination of phage predation within mucin, an important factor in the gut environment is motility-induced agitation. In experiments by Barr et al. [26,63], where the affinity of T4 for mucin was detailed, phage predation within mucin was carried out under constant flow in a microfluidic device. This flow mimicked the flushing motility within the gastrointestinal system where food and associated material, including loosened intestinal mucin and its contents, are moved down the gastrointestinal tract such as that seen during phase III of interdigestive motility. However, there are other types of agitation that promote digestion [61] such as noncyclical pattern of fed motility where mixing allows for agitation of contents within a portion of the gastrointestinal tract. To mimic the complexities of the in vivo environment and elucidate the effects on phage effectiveness, agitation (brief and constant) was added to our in-tube assays. Findings for the two phages showed that at low MOI, predation by both phages was unaffected by agitation in low mucin concentration samples (≤1%) while high mucin concentration (≥2%) with brief agitation significantly decreased predation. At high MOI, constant agitation corrected for decreased predation by T4 while high MOI of PhiX completely overcame any effects due to agitation. These data suggest that the impact of the variables influencing bacterial viability following phage exposure within the small intestine are phage dependent. 

Although there are limitations to our in vitro model system, the results speak to the nature of short-term phage–bacteria interactions in the gut within a mucin-containing environment. In vivo, bacteria entering the intestine are likely to become coated with mucin through interactions with mucin glycoproteins and assisted by the contractions of the GI tract [24]. Brief agitation might allow for full coating where more extended periods of agitation might serve as disruptive contractions to break mucin strands exposing previously unavailable regions on the bacterial surface [69]. Our findings support this as brief agitation inhibited phage predation in high mucin concentrations. It is possible that mucin affinities of the phages [9] could also affect phage predation outcomes where, at low MOI, freely diffusing phage (such as PhiX) would be unable to access bacteria for any significant interaction and phages with mucin affinity (such as T4) would also be tethered without access. As the bacteria become fully coated, they would be unable to extricate themselves from their encapsulated micro-environments and this lack of motility would further hinder interactions with phage, thus reducing effective predation. In our system, increasing numbers of phage PhiX allowed for the recovery of phage predation under all conditions tested, suggesting that increased numbers of freely diffusing phage can find the “holes” in the mucin coatings to overcome hinderance to predation. However, the less diffusible phage T4 remained hindered, suggesting that its interaction with mucin still prevented effective predation. These results may explain the bacterial overgrowth seen in conditions with decreased GI motility as in Scleroderma and Cystic Fibrosis, where increased mucin production may lead to reduced phage predation resulting in expansion of the bacterial population [32,33,70,71].

Increased agitation would allow for a shift of the system dynamics where the encapsulated mucin micro-environment would be broken apart [69,72,73]. Then, increased GI contractions might serve to provide the phage with higher affinity for mucin (T4) a way to increase the probability of encountering host bacteria that was previously “hidden” in the mucin as the agitation would allow for the movement of phage into the proximity of bacteria stuck within the mucin matrix. These phenomena could also be explained by increased bacterial motility where flagellar motion would work to “drag” phage-bound mucin strands towards the bacterial cell as proposed by Joiner et al. [51]. The fact that high MOI was still required to restore phage predation in higher mucin concentration indicates that the strongest driver of effective phage predation was the availability of more phage. 

While the in vitro systems are informative for the specific coliphage interactions tested here, in vivo experiments to test the effectiveness of phage predation within small intestinal mucin as a function of GI motility and mucin concentration are necessary. Furthermore, while both phages in this study were coliphage, they exhibit physiological differences that might also affect outcomes described herein. In addition to having no known mucin affinity, PhiX is physically a much smaller phage, which may confer inherent advantages. Namely, if another *Microviridae* phage had mucin adherent properties, it might still function similarly to or better than PhiX as its accessibility to smaller regions would also exist. Regardless, these studies offer an intriguing explanation as to how free lytic phage and their host bacteria could co-exist within a system and how minor shifts in that system (increased mucin production or decreased GI motility) might result in an uncontrolled expansion of the bacterial component.

## Figures and Tables

**Figure 1 microorganisms-11-00508-f001:**
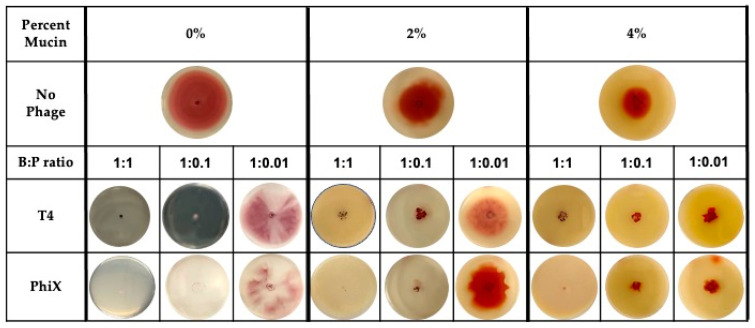
Even distribution of phages within mucin-containing motility plates allows for hinderance of bacterial motility at low MOI. Bacterial motility, as indicated by distance of bacterial metabolism (red color) from original inoculation site (center of plate) within plates containing no (0%), 2%, or 4% mucin. Plates contained no phages (**top row**), bacteria:phage (B:P) ratio of 1:1 (left columns for each percentage), 1:0.1 (center columns for each percentage), or 1:0.01 (right columns for each percentage) of either T4 (**middle row**) or PhiX (**bottom row**). Images are representative of all results (*n* ≥ 3).

**Figure 2 microorganisms-11-00508-f002:**
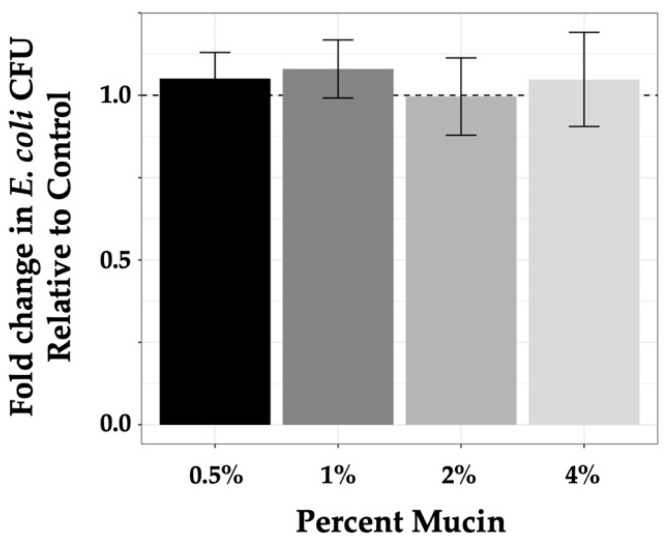
Mucin does not enhance K803 growth. Growth of K803 in increasing mucin concentrations after 3 h graphed as compared with growth of bacteria in LB alone for the same time period. Controls were normalized to 1 and are indicated by the dashed line. Bars are means with 95% CI, *n* = 3.

**Figure 3 microorganisms-11-00508-f003:**
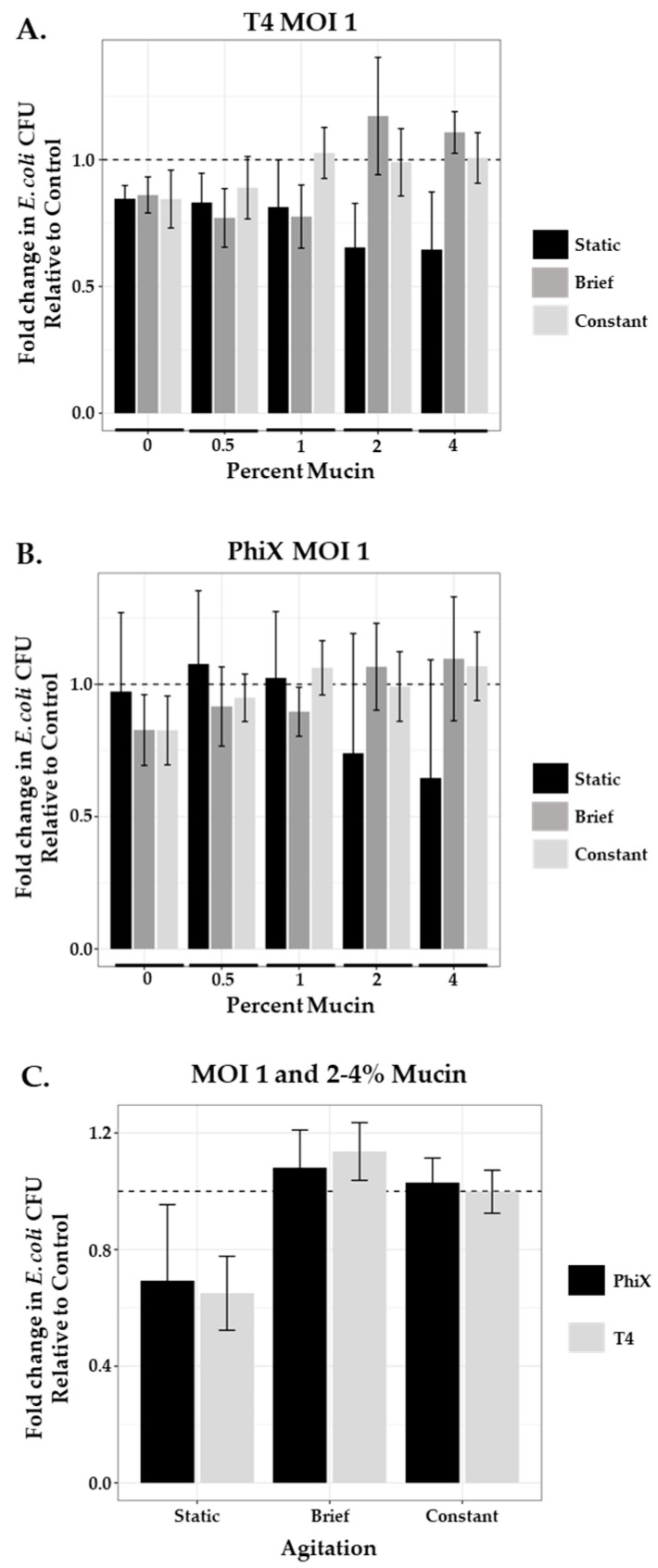
Agitation influences phage predation at high mucin concentration. Comparison of the fold change in bacterial growth as compared with controls (normalized to 1, dashed line) in static (black bars), briefly agitated (dark gray bars), and constantly agitated (light gray bars) in-tube samples of varying mucin concentration (0–4%) treated with MOI 1 of (**A**) T4 or (**B**) PhiX. Columns are means with 95% CI. (**C**) Comparison of the combined 2% and 4% mucin means of either PhiX-(black bars) or T4-(gray bars) treated samples after each type of agitation (static, brief, or constant). N ≥ 6 for all experiments.

**Figure 4 microorganisms-11-00508-f004:**
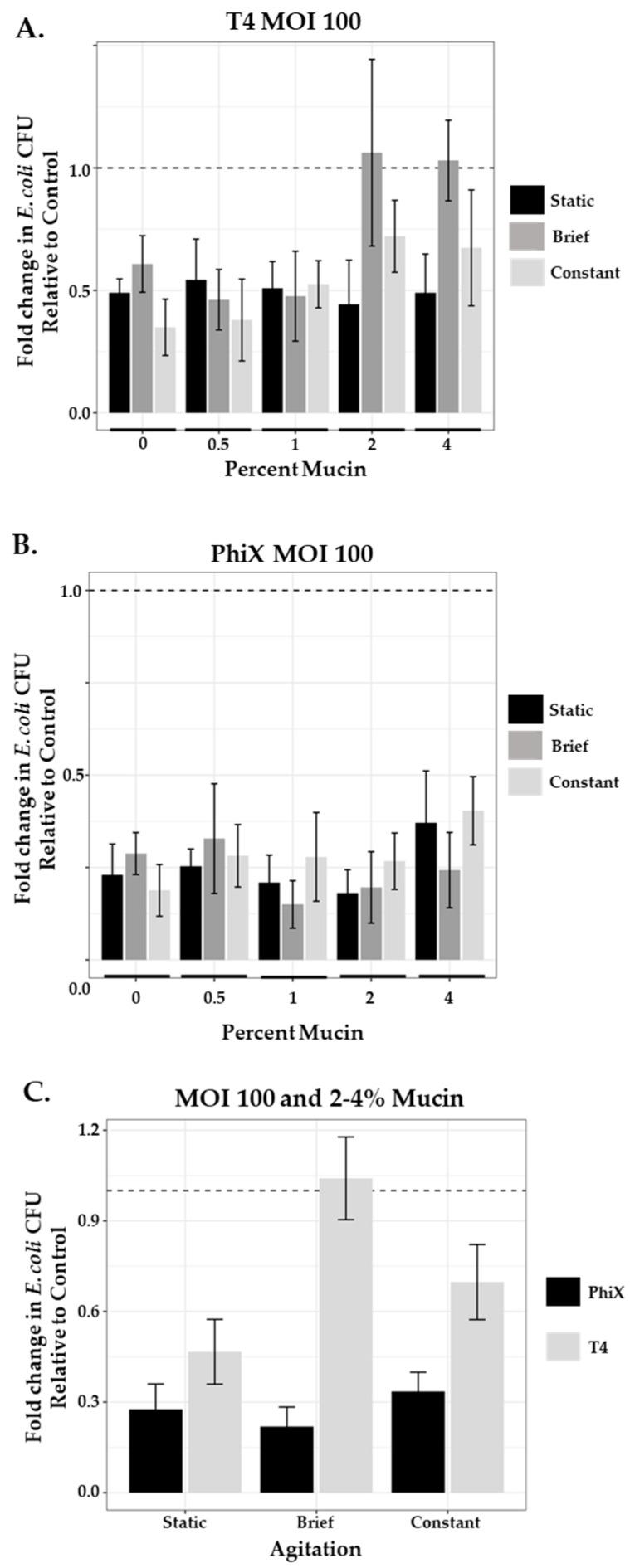
Phage predation with increased MOI of 100 is unique to each phage. Comparison of fold change in bacterial growth in static (black bars), briefly agitated (dark gray bars), and constantly agitated (light gray bars) in-tube samples of varying mucin concentration (0%–4%) treated with MOI 100 of either (**A**) T4 or (**B**) PhiX. (**C**) Comparison of combined 2% and 4% PhiX (black bars) and T4 (gray bars) means as a function of agitation. Columns (means with 95% CI) are representative of the fold change in *E. coli* growth relative to controls (normalized to 1, represented by dashed line). N ≥ 6 for all experiments.

**Figure 5 microorganisms-11-00508-f005:**
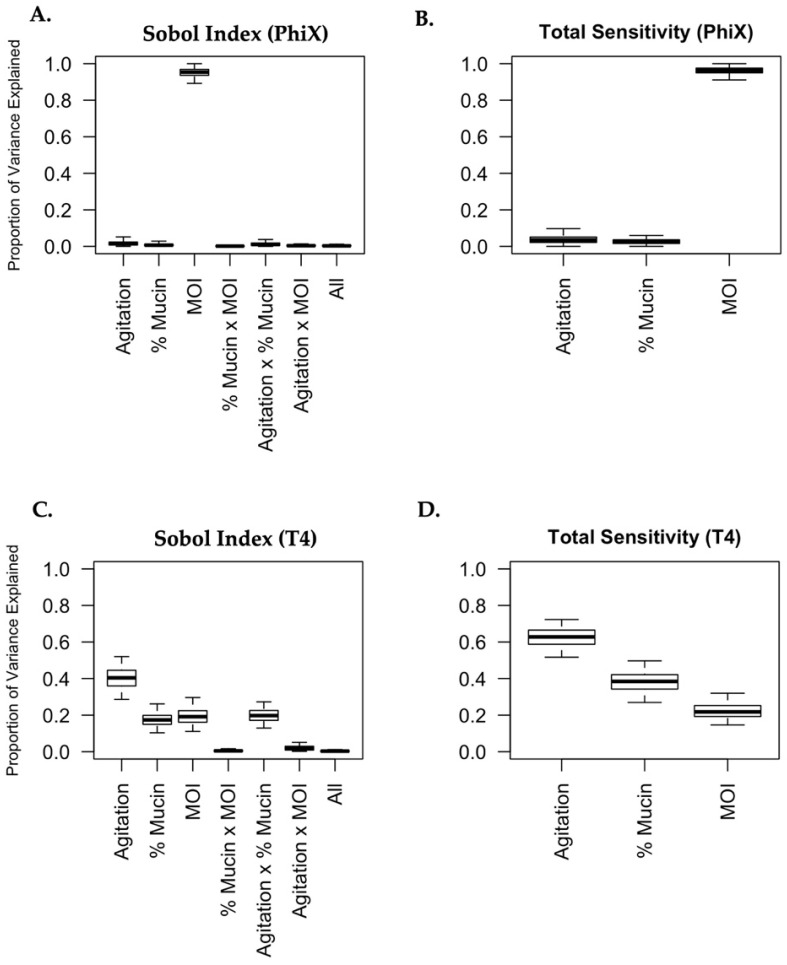
Phage predation is uniquely influenced by its surrounding micro-environment. Sobol variance-based sensitivity analyses were performed for PhiX (**A**,**B**) and T4 (**C**,**D**). The effect of each variable (agitation, mucin concentration, and MOI) and the possible interactions with other variables were broken into the proportions with which they contributed to the overall change in phage predation. The proportions were graphed as “Proportion of Variance” (**A**,**C**). The overall contribution of each variable to phage predation was also graphed as “Total Sensitivity” (**B**,**D**).

**Table 1 microorganisms-11-00508-t001:** Motile distance (mm) of *E. coli* in mucin from central inoculation point.

% Mucin	0%	0.5%	1%	2%	4%
Bacterial motility (SEM) ^1^	23.87 (0.82)	18.62 (1.34)	17.54 (0.74)	11.99 (0.87)	8.66 (1.3)
Dunnett’s *p*-value ^2^	NA	0.046	<0.0001	<0.0001	<0.0001

^1^ Standard error of the mean; ^2^ Comparison between 0% and each concentration mucin.

**Table 2 microorganisms-11-00508-t002:** Fold change in *E. coli* growth in mucin as compared with growth in Luria Bertani broth.

% Mucin	0.5%	1%	2%	4%
Fold Change in growth (SEM)	1.05 (0.04)	1.08 (0.04)	0.996 (0.05)	1.05 (0.06)

**Table 3 microorganisms-11-00508-t003:** Fold change in *E. coli* growth relative to control in static system with phage (MOI 1).

% Mucin	0%	0.5%	1%	2%	4%
T4 mean (SEM)	0.846 (0.02)	0.832 (0.05)	0.813 (0.08)	0.654 (0.08)	0.646 (0.1)
PhiX mean (SEM)	0.973 (0.12)	1.077 (0.11)	1.024 (0.1)	0.740 (0.18)	0.646 (0.17)

**Table 4 microorganisms-11-00508-t004:** Experimental means of phage containing 2% and 4% mucin samples at MOI 1.

	Static Mean (SEM)	Brief Mean (SEM)	Constant Mean (SEM)	Tukey’s *p*-Value ^1^ (Brief/Constant)
T4 2% mucin	0.654 (0.08)	1.172 (0.1)	0.990 (0.06)	0.003/<0.0001
T4 4% mucin	0.646 (0.1)	1.108 (0.04)	1.007 (0.04)	0.012/0.002
PhiX 2% mucin	0.740 (0.18)	1.066 (0.07)	0.991 (0.06)	0.02/0.08
PhiX 4% mucin	0.646 (0.17)	1.096 (0.1)	1.068 (0.06)	0.02/0.04

^1^ Tukey’s multiple comparisons test with static at same mucin concentration.

**Table 5 microorganisms-11-00508-t005:** Fold change in *E. coli* growth relative to control in static system with phage (MOI 100).

% Mucin	0%	0.5%	1%	2%	4%
T4 mean (SEM)	0.490 (0.03)	0.543 (0.07)	0.509 (0.05)	0.443 (0.08)	0.490 (0.07)
PhiX mean (SEM)	0.230 (0.04)	0.253 (0.02)	0.209 (0.04)	0.181 (0.03)	0.371 (0.06)

**Table 6 microorganisms-11-00508-t006:** Means of phage containing 2% and 4% mucin samples at MOI 100.

	Static Mean (SEM)	Brief Mean (SEM)	Constant Mean (SEM)	Tukey’s *p*-Value ^1^ (Brief/Constant)
T4 2% mucin	0.443 (0.08)	1.062 (0.14)	0.721 (0.06)	0.015/0.004
T4 4% mucin	0.490 (0.07)	1.030 (0.07)	0.674 (0.1)	0.001/0.016
PhiX 2% mucin	0.181 (0.03)	0.196 (0.04)	0.267 (0.03)	0.951/0.234
PhiX 4% mucin	0.371 (0.06)	0.243 (0.05)	0.404 (0.04)	0.174/0.89

^1^ Tukey’s multiple comparisons test with static at same mucin concentration.

**Table 7 microorganisms-11-00508-t007:** Estimated total sensitivity scores for each experimental factor.

	Agitation Level		% Mucin		MOI	
	Estimate	95% CI ^1^	Estimate	95% CI ^1^	Estimate	95% CI ^1^
PhiX	0.038	(0.00, 0.10)	0.027	(0.00, 0.06)	0.961	(0.91, 1.00)
T4	0.625	(0.52, 0.72)	0.383	(0.27, 0.50)	0.223	(0.15, 0.32)

^1^ 95% credible intervals.

## Data Availability

Not applicable.

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
