# Peer review of "Mucin and Agitation Shape Predation of *Escherichia coli* by Lytic Coliphage"

_microorganisms, 2023, doi:10.3390/microorganisms11020508_

Round 1

Reviewer 1 Report

The manuscript describes an analysis of the effect of mucin on the number of surviving bacteria after phage attack under various conditions. Two different phages, T4 and PhiX were analyzed with strain K803 as host.

·         Figure 1 and Table 1 shows results from an experiment, where bacterial swarming in soft agar from an application spot in the center of the plate, with varying concentrations of mucin, was monitored. Red color development from a tetrazolium dye by acidification of the medium was used as a visualization of bacterial survival and extent of swarming. The effect of phages at different concentrations was analyzed by including phages in the soft agar. Without addition of phages in the medium, the spread of red color from the central application spot was limited by 2% and 4% mucin, showing that mucin limited bacterial swarming. Without mucin, phages at low concentration limited bacterial spreading to few resistant cell lines. However at higher concentrations phages prevented red color formation. At high mucin concentrations, spreading was severely restricted to small red circles, which appeared to contain bacteria that was resistant to phages.

o   The response of the bacteria in the experiment is very interesting, but the use of MOI quantification makes no sense. In the bacterial application spot the actual MOI is likely to be less than one percent of the reported MOI. As the cells swarm and grow, the actual MOI is not easy to quantify. The phage concentration should be given as pfu/ml.

·         Simple growth experiments showed (Figure 2 and Table 2) that mucin had no effect on the number of cfu from K803 cultures. For the growth experiments, tubes were inoculated with 1000 bacteria from diluted stationary phase cultures in LB, which had been left at room temperature for 30 min. Following growth at 37 oC for 3 hours, aliquots were plated on agar plates (presumably LB plates) and after overnight incubation, plates were photographed and a defined area was used for computer aided colony counting. The result of this quantification was reported as relative cfu, compared to samples without mucin.

o   This experiment is problematic for several reasons. Stationary cultures of bacteria may show a large lag phase, so the cells may not have started growing before after 1 or 2 hours. The 3 hour growth will not be enough to let the cells change from preferred nutrients to mucin. The cfu counts of the control before and after the 3 hour incubation should be reported as a minimum, to show if the cells started growing. A discussion of why this growth assay was employed should be included.

·         Analysis of phage effect on the growing K803 was analyzed using the same growth assay, but the incubation phase was defined as Static, gentle and Constant. If I understood it right,

o   Static cultures contained a mixture of 1 ml LB medium, mucin and phages in 1.5 ml (Eppendorf ?) tubes. Then 0.01 ml Stationary phase bacteria at room temperature were added without mixing. After 3 hour incubation (without mixing ?) 0.1 ml was withdrawn without mixing and plated for cfu determination. If this procedure should be reproduced, it contains so many unknowns that the current description is inadequate.

o   Brief (=gentle?) cultures in Eppendorf tubes were mixed by single inversion of the tube after addition of bacteria, once after addition of phages, and once before taking the sample for cfu. Else they were left for static incubation.

o   Constant cultures were incubated with shaking at 100 rpm.

o   In the static system, the effect of phage at a MOI of 1 (1000 pfu per tube?) appears to result in a reduction of bacterial cfu of only 15% (Table 3). This could be taken as evidence that something is missing from the phage killing assay, as phages would normally kill off the majority of the bacteria. The low bacterial and phage concentration (1000/ml) would of course lower the probability of infection, but the killing appears insignificant. The same is seen for the rest of the experiments, so it appears as if the assay is not sensitive enough.

o   If the assay, for comparison with the gut microbiome, has to contain the low concentration of bacteria, phage killing has to be reported with higher concentrations first, to show that the growth conditions can result in severe bacterial killing. The design of the assay and the growth conditions has to be discussed and rationalized.  

 The topic is very interesting, but if it is true that the assay leads to more than 10% surviving bacteria, a re-design of the experiment can be needed. 

Reviewer 2 Report

The submitted manuscript by Carroll-Portillo et al, describes how mucine and agitaion have an influence on E. coli predation by lytic coliphages. The manuscript is well written with meticulously carried out experiments. The initiative is also interesting as it is aims to through more light on the factors influencing phage dependent lysis to combat bacterial infection in GI tract. However, all the experiments have been carried out solely in relative simple "in vitro" background which should simulate the GI tract environment and conclusions have been drawn. For initial understanding of the interactions (mucin, agiation, E.coli and coli phages) the ease of the set up is considerable. However, as the microcosmos of GI sytem is much more complex with multiple interactions, I doubt whether the conclusions drawn from the findings of this manuscript are very meaningful and could be universally applied. Interestingly, in a review article from Joana Costa 2019, different classes of bioreactors and flow systems and some of their technical specifications have been described which according to me would have been be a better option to select one such and use it as Intestinal in vitro Model to test the influencing factors. Moreover, in the introduction general information about mucin and its role to protect GI mucosa is lacking. I think, from the same authour there has already been a review (Carroll-Portillo 2021) which adresses such issue but it is missing in the reference of this current manuscript. Line 140: "Constant agitation involved incubating samples in a 37°C shaker at 100 rpm ":  I personally find this speed to high; could be around 50 rpm when it should simulate human GI tract movements. In Figure 3, use further labels as A, B and C to make it convenient for understanding. 

Reviewer 3 Report

File attached

Author Response

We would like to thank this reviewer for their guidance and believe that our paper is stronger due to changes made in response to comments. Please see our responses below:

  • English corrections are required.
    • Done
  • The main methods should be clear enough for the readers to replicate in their labs.
    • Done. We have rewritten for clarity.
  • For plate motility assays, authors have evaluated MOI 1, .1 and .01 whereas for tube assays they evaluated 1 and 100. What’s the rationale?
    • The justification for our choice of MOI 1 and 100 is described in lines 364-369 and 438-442. As in-tube assays with MOI 0.1 did not demonstrate any measurable bacterial killing within a three hour time frame,  we tested MOI of 1. MOI of 100 was selected to be representative of concentrations associated with phage therapy application.

  • Eppendorf is a brand; please replace it with “microfuge or microcentrifuge tubes”.
    • Done
  • At some places authors have used “brief” and other places they used “gentle”. Please maintain the uniformity.

    • Done
  • Instead of fold changes in the viable cell counts of the bacterial populations. CFU/mL.
    • The method of analysis chosen for determination of bacterial viability on the plates was to remove bias introduced by colony counting methodologies. Plates from samples treated with phage often had minute or “smearing” of bacteria. These morphological differences that were visually obvious would be missed using CFU as small bacterial colonies are equivalent to large, healthy colonies using this metric. As such, inclusion of all growth over an area was selected with fold change in that growth being the final output. We have added a supplementary figure (Supplementary Figure 2) along with text to the manuscript to further explain this decision. See lines 202-204 in Methods and lines 349-359.

Reviewer 4 Report

The part of discussion need simplify, please revise.

Author Response

We would like to thank this reviewer for contributing their time and comments to this manuscript. We feel our paper has been made stronger by all the reviewers' suggestions. Please see our response below.

  • Discussion need to simplify
    • Done

discussion needs to simplify\D

discussion needs to simplify\

Round 2

Reviewer 1 Report

The points raised about the description of MOI and agitation have been met. 

However to me, the method of analysis is still not justified.

The effect on the survival of the bacteria after phage attack is extremely small in this study. Raw data should, as a minimum, be supplied for the control strain to show that a substantial fraction of the bacteria had been killed during the three hours of incubation.

It is very unusual that survival during phage attack can be replicated within 20% error, when less than 0.1% of the bacteria have survived.

Author Response

We thank this reviewer for their continued guidance and believe that our paper is stronger due to the changes we have made in response to the comments. Please see our responses below.

  • The effect on the survival of the bacteria after phage attack is extremely small in this study. 
    • Even though the effects seen were small, they are consistent over a large number assays (at least 9 for each condition [static, brief, constant] with triplicates each time). We have included a new supplementary figure (Supplementary Figure 2) to show representative images of the bacterial outgrowth plates to demonstrate the colony formation that occurs after a 3 hr incubation with either MOI 1 or 100 of each phage. As is demonstrated in the figures, in LB, substantial killing only occurs with MOI 100 of either phage (Found as “0% Mucin” in Figures 3A & B and 4A &B). We have also uploaded the raw data for a majority of the plate counts to demonstrate this killing by both phages at MOI 100 (represented by LB + MOI 100 rows under each condition).
  • It is very unusual that survival during phage attack can be replicated within 20% error, when less than 0.1% of the bacteria have survived.

    In our experiments, we see a wide range of "Fold change" values (ranging from 0.15 to 1.1), none of which are compatible with the reviewers claim. Unless the reviewer is able to clarify their comment, we are unable to address this further.

Reviewer 2 Report

Format the Tables 1, 5 and 7 respectively in a similar way like the other Tables of the Manuscript.

Author Response

We would like to thank this reviewer for their previous comments as well as their current suggestions. We feel that our paper is stronger because of their contribution. Please see our response below:

  • Format the Tables 1, 5 and 7 respectively in a similar way like the other Tables of the Manuscript.
    • Done